# School Gardening, Cooking and Sports Participation Intervention to Improve Fruits and Vegetables Intake and Moderate-to-Vigorous Physical Activity among Chinese Children: Study Protocol for a Cluster Randomized Controlled Trial

**DOI:** 10.3390/ijerph192114096

**Published:** 2022-10-28

**Authors:** Yufei Qi, Siyu Rong, Kunlong Liao, Jiaqi Huo, Qian Lin, Sareena Hanim Hamzah

**Affiliations:** 1Faculty of Sports and Exercise Science, University of Malaya, Kuala Lumpur 50603, Malaysia; 2Department of Physical Education and Research, Central South University, 932 Lushan south Rd., Changsha 410083, China; 3Zhangshumen Primary School, Halfway Street, Taohualing Village, Changsha 430100, China; 4Department of Nutrition Science and Food Hygiene, Xiangya School of Public Health, Central South University, 110 Xiangya Rd., Changsha 410078, China

**Keywords:** gardening intervention, sports participation, fruits and vegetables, rope skipping, school children

## Abstract

Inadequate intake of fruits and vegetables (FV) and moderate-to-vigorous physical activity (MVPA) in children has become a global public health problem. Therefore, school-based gardening and cooking (SGC) and sports participation (SP) interventions may be effective in improving children’s FV intake and MVPA. The aim of this study is to develop and evaluate the effectiveness of SGC and SP interventions on FV intake and MVPA among Chinese children. In this cluster randomized controlled trial study, 237 children in grades 4–5 from six public primary schools from Changsha, Hunan Province, China will be randomly assigned to: (1) a SGC and SP combined intervention group; (2) a SP intervention group; (3) a regular practice group. The intervention clusters will be implemented for a period of 6 months and follow up will be carried out after 12 months. The outcome will be collected using a combination of self-reported and objective measures. Primary outcomes will include children’s FV intake and duration of MVPA per day, and secondary outcomes will included frequency and attitudes of FV intake and SP, in addition to other measures. Finally, a process evaluation will be used to analyze the facilitators and barriers to intervention implementation. Trial Registration: (Registration Number: ChiCTR2200064141).

## 1. Introduction

An inadequate intake of fruits and vegetables (FV) and insufficient physical activity (PA) among children have become global health concerns [1,2,3]. Inadequate intake of FV in children not only leads to deficiencies in vitamins, minerals and dietary fiber, which are essential to human health [4,5], but it also increases the risk of many chronic non-communicable diseases in adulthood, such as obesity, diabetes, cardiovascular disease, cancer, etc. [6,7,8,9]. Additionally, insufficient moderate to vigorous physical activity (MVPA) is associated with an increased risk of obesity, metabolic syndrome and mental health problems in children [10,11,12].

Although many studies have shown that dietary and PA interventions can improve FV intake and MVPA in children [13,14], and in particular that multifaceted interventions using both diet and sports participation (SP) are more effective than diet or SP interventions alone in improving children’s health [15,16,17], there is still a lack of sustainable and structured intervention programs. Most of the current diet and SP program for children in China have been designed with the specific goal of improving obesity, neglecting the development of lifestyle-related interventions such as improving children’s eating and PA behavior [18,19,20]. However, the development of good lifestyles during childhood is important in promoting health and preventing disease [21,22]. In recent years, there has been increasing interest in the application of intervention programs for school gardening and cooking (SGC) activities combined with SP to improve children’s eating and PA behavior [16,23,24].

SGC activities is a school-based nutrition intervention project; by planting and cooking FV in schools, students can receive nutrition knowledge under the guidance of teachers and experience the process of planting and cooking FV [25]. SP is a subset of PA and youth sport has been defined as a program that provides a systematic sequence of practices and contests for children and youth [26]. Many studies show that SGC activities are effective in improving children’s FV intake nowadays [27,28,29], and that different forms of SP can have a significant effect on improving children’s MVPA (especially sports that are both fun and challenging to ensure adherence to the intervention) [30,31,32,33]. However, research in this area is still very limited in China. For children, most of their waking hours is spent at school, making it the best environment for implementing SGC activities and SP interventions [34]. In China, the *Curriculum Program and Curriculum Standards for Compulsory Education (2022 Edition)*, published by the Ministry of Education in 2022, further optimizes the curriculum, such as the complete separation of labor from the former integrated practical activities curriculum and the publication of *the Compulsory Education Labour Curriculum Standards (2022 Edition).* Schools are required to organize students to participate in daily life work, productive work and service work in a planned manner, with an average of no less than one lesson per week. The focus is on requiring primary school students in grades three to four to experience FV growing and to learn to prepare FV salad, while primary school students in grades five to six are expected to grow common local FV trees and to learn to cook two to three home-cooked dishes. In addition, the *Physical Education and Health Curriculum Standards for Compulsory Education (2022 Edition)* shows that the percentage of “Physical Education and Health” has increased to 10–11%, making it the third most important subject, and requires the integration of new sports such as fancy jumping rope into the teaching content. Therefore, implementing a structured school-based intervention program for children based on SGC and SP aligns with curriculum requirements in China.

A recent study used the multifaceted intervention effectively reduced the mean BMI and obesity prevalence in primary school children across socioeconomically distinct regions in China [35], but it did not change MVPA, physical fitness, and other outcomes. This is mainly due to the fact that the researchers did not use sport participation interventions that were fun and challenging. Instead, students are required to do PA for a total of one hour per day, which does not allow for a good level of intensity and duration of PA for children. Therefore, the choice of a sensible intervention is crucial. Based on the latest education policy in China, a physical and nutritional health promotion program for primary school children in grades four to six will be developed using a combined SGC and SP intervention to increase the intake of FV and the duration of MVPA among school children aged nine to twelve years, and to improve their FV intake and SP frequency, attitude and preference, with the following specific objectives: (1) to develop a program that included sports participation in the existing SGC intervention program for primary school children in China; (2) and to explore the effects of SGC and SP on the intake of FV, intake motivation, time and frequency of MVPA, and other healthy conditions among school children in China. 

## 2. Materials and Methods

### 2.1. Study Design

This protocol has been written in accordance with the Recommendations for Interventional Trials statement and is a cluster randomized controlled trial [36]. There are a total of six main urban districts in Changsha, Hunan Province, China, and we will use a two-stage cluster sampling method to randomly select six public primary schools from each of the six districts. All school-age children in grades four to five in each school will be included in this study. Two classes from grades four to five in each school will be selected using a random group selection process, and all students in the class will be invited and voluntarily enrolled in this study. After completion of the baseline survey, the six public primary schools will be randomized in a 1:1:1 ratio into (1) a SGC and SP group; (2) a SP intervention group; and (3) a regular practice group. All groups will receive 45 min of intervention four times a week. A final school year of intervention and follow-up will be implemented for each group, with effects assessed at 6 and 12 months. An overview of the study design is presented in Figure 1.

### 2.2. Recruitment

#### 2.2.1. Recruitment of the Schools

Considering the limited cognitive ability of grade three school-age children who cannot guarantee accurate completion of self-reporting and the fact that grade six school-age children will soon move on to junior high school and have limited time at school, which may have some impact on subsequent follow-up, we have chosen grade four to five school-age children as the target group for recruitment, not only so that they have good receptivity to educational information but also so that they have sufficient time to complete the intervention and follow up. The schools will be randomly selected if they meet the criteria described below:

Inclusion criteria: Schools will be included if they were public primary schools and with a minimum of 300 primary students. The school principal agrees to the randomization process and comply with this study protocol.

Exclusion criteria: Schools will be excluded if they are boarding schools and schools for special populations, as are schools with fewer than two classes in grades four to five or schools with plans to cease operating within a year, and schools without established gardens or sufficient space to plan garden construction, or receiving other similar interventions will be excluded.

#### 2.2.2. Recruitment of the School Children

Informed consent forms will be provided to all students and their primary caregivers after school recruitment is completed and before the baseline survey begin. A questionnaire on the student’s health status will be distributed to each student and primary caregiver after they had both signed informed consents. The project staff will collect the questionnaires and if a parent reports one of the following conditions, his or her children will be excluded: (1) hypertension, medical history of heart disease, asthma, diabetes, tuberculosis, hepatitis, or nephritis; (2) obesity caused by side effects of drugs or endocrine diseases; (3) abnormal physical development such as gigantism or dwarfism; (4) physical deformity such as severe scoliosis, limp, pectus carinatum, obvious X-leg or O-leg; (5) unable to participate in moderate to vigorous physical activity; (6) unable to understand the information in the questionnaire. Ultimately, we will recruit 96 primary school students in grades 4–5 in each school. If the total number of students in two classes is not 96, we will randomly select a small number of students from grade three until the sample size is reached.

### 2.3. Sample Size Estimation

With PA duration rate (PA duration > 60 min per day) and FV intake rate (FV intake >700 g per day) as a calculated basis: control group weekly PA and FV intake rate divided into p1 = 20.6% and p1 = 23.1%, it is expected that after the intervention the daily PA hours and FV intake of the intervention group is 50% of the standard, other control groups are not considered for the time being, an alpha of 0.05, power of 90% and 20% of lost visits are considered, the final settlement results: 79 people in each group, a total of 237 people.

### 2.4. Randomisation Procedures

Schools located in six different regions (clusters) will be randomly assigned to the combined intervention group, the single intervention group or the control group using a computer-generated random number system (simple random sampling method) in a 1:1:1 ratio. Randomization will be carried out by an independent staff member from the Department of Nutrition and Food Hygiene, School of Public Health, CSU. To ensure allocation concealment, randomization will not be carried out until after the baseline survey has been completed. Blinding of the school allocation in this setting will not be possible. Therefore, the investigators and data collectors will not be blinded in this trial. However, the outcome assessors and the data analysts will be blinded.

### 2.5. Intervention Selection and Theoretical Basis

The SGC and SP courses are adapted from the *Junior Master Gardener Health and Nutrition* from the *Garden and the Walk Across Texas programs*, respectively. The program was developed by Texas A&M AgriLife Extension Service and has a well-established curriculum [37]. This academic year intervention is based on social cognitive theory [38], as shown in Figure 2, with the primary goal of engaging students at school and at home to promote behavioral change in diet and PA through personal and environmental influences. All programs will then be adjusted and modified based on pre-experimental results and focus group discussions.

To fill the research gap, the intervention was developed in four stages in line with the new policy on school education in China. (1) We have systematically reviewed the previous literature to identify measures relevant to the form and effectiveness of this intervention [24]. (2) Focus group discussions and interviews will be conducted with key informants (children, parents, teachers, head teachers, local health and education officials) to further revise and refine the intervention program. (3) We will conduct a 3-month pilot study with 90 grades 4 students (mean age: 9.38 ± 0.62 years) in a primary school in Changsha to test the feasibility of the initial proposed intervention. (4) After many rounds of discussions with nutrition experts, sports and health experts, school leaders and teachers, we will adapt the curriculum to suit the teaching of Chinese primary school students.

### 2.6. Description of the Intervention

The intervention consisted of two curriculum components, SGC intervention and SP intervention, with the SGC intervention curriculum consisting of three modules of practical gardening intervention, theoretical gardening and nutritional intervention and culinary intervention, and the SP intervention curriculum consisting of two modules of practical motor skills intervention and theoretical exercise health curriculum, with information about the specific intervention components shown in Table 1.

#### 2.6.1. SGC Intervention

Members of the project team received training from the Texas A&M AgriLife Extension Service and received Proof of completion certification and extension authorization. The SGC intervention was adapted from the Junior Master Gardener Health and Nutrition from the Garden, retaining the core content of the original curriculum, but with localized innovations based on Chinese schooling policies, traditions and culture, and climate. For example, in Chinese primary schools, class sizes are required to be no more than 45 students per class. Class sizes in various primary schools in the main city of Changsha are around 45–50 students, far larger than class sizes in American primary schools, so changes have to be made to suit the actual situation when designing practical gardening interventions, classroom games and some other interactive sessions. The 24 solar terms are traditional Chinese culture and play an important role in guiding people in their agricultural activities and dietary intake. For example, planting specific types of FV in specific seasons not only has a high survival rate but also yields a large amount of fruit; the intake of specific types of FV in specific seasons can bring various health benefits to the human body, so this component can be added to the curriculum design. Also, in the spring and autumn in Changsha there is a lot of rain and measures should be considered to ensure that the practical horticultural interventions run smoothly during these seasons.

#### 2.6.2. SP Intervention 

The SP intervention is based on the Walk Across Texas program and has been adapted to Chinese school policy while maintaining the core content of the curriculum. In Changsha, schools are required to implement the ‘one hour of school exercise per day’ policy and to instruct students in the skills of emerging sports, so rope skipping as an entrance exam item has become the main form of SP intervention since it is (1) not restricted by space and equipment and is easy to do in schools; (2) both fun and challenging and is conducive; and (3) the intensity of the exercise meets the requirements of MVPA, which has more potential benefits for children physical and mental health. Therefore, rope skipping has been identified as the main form of intervention, and on top of this, some sports competitions and fun games will be incorporated, with prizes and certificates to be awarded to students who achieved their goals or rankings, thus attracting more students to participate in the long term. A sports health course will also be developed, focusing on an overview of sports and emergency management of sports injuries, so that students could enrich their knowledge base while bringing knowledge transfer and first aid help to themselves and to them.

#### 2.6.3. Regular Practice Group

School children in the regular practice group will continue to follow their regular teaching schedule during the study without exposure to any of the intervention components of this study. Immediately after completing one academic year of intervention and follow-up for the intervention group, the control group will receive the same materials related to the intervention components as the intervention group.

### 2.7. Process Evaluation

Process evaluation is important in explaining the reasons for the discrepancies between expected and observed outcomes and in designing and implementing more rational and rigorous intervention program for the future [39]. Throughout the intervention, we will identify the process evaluation elements and procedure based on the principles and steps described in the conceptual framework by Saunders et al. [40], including: (1) frequency and intensity of interventions actually delivered by teachers, volunteers, program team members (dose delivered); (2) student and primary caregivers class attendance will be recorded at the schools to establish exposure and participation in the intervention, logbooks will be recorded to understand how satisfied they are with the intervention (dose received); (3) classroom observations and surveys of teachers, volunteers and program team members to assess the extent to which the intervention is being implemented as intended (fidelity); (4) interviews with school principals, teachers, and volunteers to understand the level of support from schools for the program (context).

### 2.8. Outcome Evaluation

The outcome indicator measure will be administered at three points in time. Firstly, a baseline measurement will be taken prior to randomization to a group of six public primary schools. Then the same measurement will be taken after the completion of the six-month intervention, and a final identical measurement will be taken six months after the completion of the formal intervention period to assess the sustainability of the intervention (6 months after completion of the formal intervention period, as recommended by the National Obesity Observatory). Data collection will be completed by trained data collection workers using paper questionnaires, exercise bracelets, height boards, scales and tape measures for students, and electronic questionnaires and interview guides for parents using those already entered into Questionnaire Star. The specific measurement information is shown in Table 2.

### 2.9. Outcomes Measure

#### 2.9.1. Primary Outcome 

Daily intake of FV (unit: g).Daily minutes of MVPA (unit: minutes).

#### 2.9.2. Secondary Outcome

The children’ FV preferences, identification and the attitude and willingness to eat FV.PA frequency and attitudes of children.Daily sedentary time, video screen time and sleep time (unit: minutes).Parental attitudes towards children’s FV intake and SP.Children’s height, weight and waistline.Questions and recommendations on the implementation of the interventions.

### 2.10. Statistical Analyses

Statistical analysis will be performed using SPSS 26.0 statistical software. According to the data characteristics, the children’s intake of FV, MVPA, and sedentary behaviors and weight status will be statistically described, and the chi-square test and the nonparametric rank-sum test will be used for univariate analysis to compare the differences in intake of FV, MVPA and sedentary behaviors and weight status in the intervention group and the control group. The generalized estimating equation (GEE) will be used to take intake of FV, MVPA and sedentary behaviors and weight status as dependent variables, and at the same time, the interaction between intervention effect and time will be considered to analyze the influence of the gardening intervention on intake of FV, MVPA and sedentary behaviors and weight status. The test level of all analytical methods is α = 0.05 (two-sided), and the difference is considered statistically significant with *p* ≤ 0.05. 

### 2.11. Cost-Effectiveness Analysis

We will assess the cost-effectiveness of the program in terms of cost per case of fruit and vegetable intake or MVPA improved. 

Cost: The cost will include the time spent by project staff, school staff and the student’s primary caregiver (in most cases the parent) for all intervention activities and material costs. Only the time spent by project staff implementing the intervention will be included. Time costs will be based on individual employment compensation (if available) or the average compensation for similar employees in the local area. The cost of garden materials, seeds, fertilizer, kitchen utensils, kettles, and hoes are subject to actual purchase prices.

Effectiveness: The impact of the program will be measured by changes in fruit and vegetable intake or MVPA. The cost-effectiveness ratio will be calculated as follows:Cost of intervention services for one primary school child.Cost for each case of improved fruit and vegetable intake.Cost for each case of improved MVPA.

### 2.12. Patient and Public Involvement

Participants (children, primary caregivers, teachers, school principals, local health and education officials) in this study will participate in interviews and focus group discussions at the time of the improvement intervention program. They are not involved in the formulation of idea development, research questions, trial design, outcome measures, recruitment or study execution, data collection and analysis and interpretation of the results. In addition, they will be invited to participate in conferences to disseminate research findings to other public schools after they have been translated into Chinese.

## 3. Discussion

Although some intervention studies have been conducted in China to improve FV intake and MVPA in children, there are still gaps in the mix of interventions and in the sustainability of interventions. This should be the first cluster randomized controlled trial of school gardening, cooking activities, and sports participation in children in China. By establishing a sustainable and structured intervention program to validate its effectiveness and sustainability in improving children’s FV intake and MVPA deficits, evidence will be provided to the education sector and policy makers for the national replication of this program. 

There are several strengths of this study. First, it will be the first cluster randomized controlled trial using a combined gardening, cooking, and sports participation intervention in China. Second, the implementation of the intervention involves not only students but also primary caregivers and teachers to increase the potential for sustainability. Third, this study will assess changes in outcomes using both self-reported and objective measures, with more rigorous results. Fourth, the study will develop a process evaluation plan for the implementation of the intervention and conduct a six-month follow-up to determine the sustainability of the effect. However, limitations of the study should be considered. First, the duration and outcomes of this study is limited and more funding will be applied to implement long-term interventions in the future. Second, FV intake were measured by self-report, which is easily affected by social recognition bias, so they may not accurately represent the changes in dietary intake, studies should consider the use of serum carotenoids as dietary biomarkers in the future.

In order to ensure the sustainability of SGC and SP interventions, there are a few things which should be considered carefully. The first thing to do is to maintain the garden, kitchen and sports facilities; it is recommended that the school set up a committee including school administrators, teachers, parents, and students to work out a plan to ensure that the garden is watered and weeded, the kitchen and sports facilities is maintained on a rotating basis during weekends (and during the summer and winter holidays). Secondly, the project leader and members of the team will train interested school teachers and staff on the topic, teaching them how to teach SGC and SP and integrate them into the existing school curriculum. Finally, outreach work could be carried out in conjunction with the local education authorities, for example by producing a web-based resource containing a syllabus for SGC and SP, implementation guidelines and teaching videos for use by other schools.

## 4. Conclusions

The intake of FV and the increase of MVPA play an important role in the development of healthy habits and normal BMI for children, however, inadequate intake of FV and insufficient MVPA in children still exist in many countries. This study intends to use a cluster randomized controlled trial to explore the effects of SGC and MVPA on the fruit and vegetable intake and MVPA of Chinese children, and provide new ideas for researchers to tackle problems such as overweight and obesity in children in the future.

## Figures and Tables

**Figure 1 ijerph-19-14096-f001:**
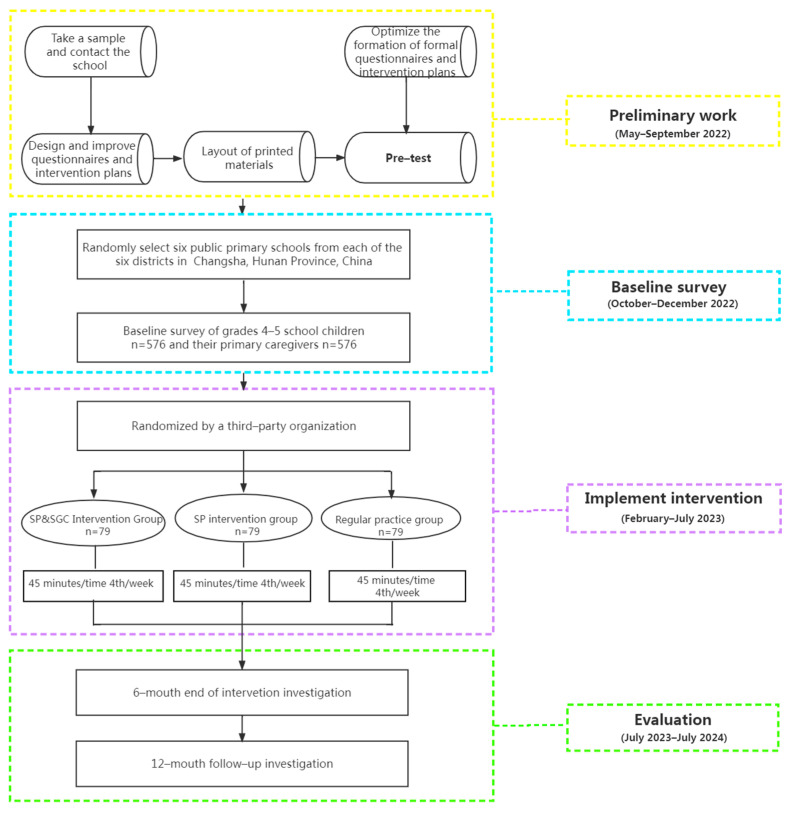
Study flow chart.

**Figure 2 ijerph-19-14096-f002:**
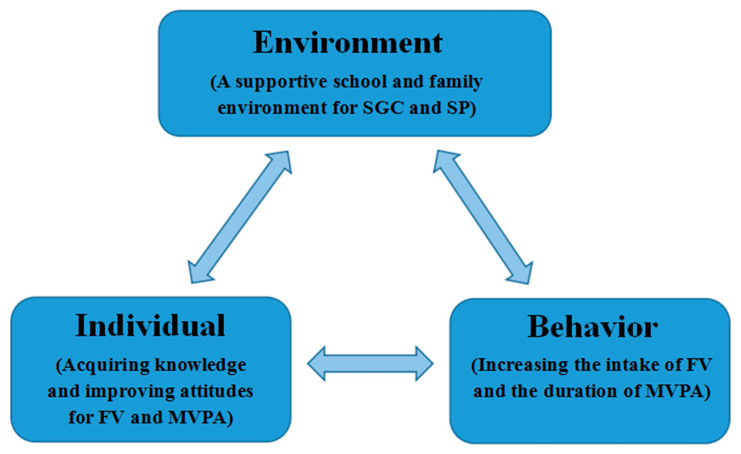
Intervention in the development process.

**Table 1 ijerph-19-14096-t001:** School gardening and cooking and sport participation intervention components.

Intervention	Intervention Components	Target	Descriptions of the Content, Frequency and Duration	Person Responsible
School Gardening & Cooking	SGC Training	Teachers	Content: SGC intervention introduction; the specific operation method of each intervention process; the teacher version of the intervention process diary and partial data collection. Frequency: Only once before the intervention. Time: 1–2 h.	Project team leader and members
	Gardening Practice Intervention	Students	Content: Planting 16 seasonal vegetables and fruits in the school garden. Among the vegetables are peppers, cucumbers, beans, aubergines, red amaranth, hollow cabbage, perilla, spinach, oleander, cabbage, string beans and lettuce. Fruits include strawberries, thumb watermelon, melon and cherry tomatoes Frequency: one intervention every one to two weeks, ten times in the last semester, and six times in the next semester. Time: Each intervention lasts for 45 min.	Teachers, volunteers, and project team members
	Gardening and Nutrition Theory Intervention	Students	Content: A total of 16 gardening theory courses, including the needs of plant growth, human body growth, the importance of vegetables and fruits, 24 four solar terms and planting, etc. Frequency: one intervention every two weeks, eight times in the last semester, and eight times in the next semester. Time: Each intervention lasts for 45 min.	Teachers, volunteers, and project team members
	Cooking Intervention	Students, Primary caregivers	Content: eight seasonal vegetable and fruit salad cooking demonstration and tasting, and share the eight vegetable and fruit recipes. Frequency: One intervention every four weeks, four times in the last semester, and four times in the next semester. Time: Each intervention lasts for 45 min.	Teachers, volunteers, and project team members
	Home Gardening Intervention	Students, Primary caregivers	Content: Students grow fruits and vegetables with their parents in simple containers at home, and share gardening and nutrition theories with their parents. Frequency: no requirement. Time: No requirement.	Teachers
	Student Diary	Students	Content: Students should take the initiative to write a small diary of about 300 words after completing the activities related to gardening practice, gardening and nutrition theory knowledge learning, and salad tasting. Frequency: Writing is required after each intervention. Time: No requirement.	Teachers
	SGC Toolkit	Teachers, Volunteers	Content: Materials, tools, and textbooks.	Teachers
Sports Participation	SP Training	Teachers	Content: SP intervention introduction; the specific operation method of each intervention process; the teacher version of the intervention process diary and some data collection. Frequency: Only once before the intervention. Time: 1–2 h.	Project team leader and members
	Sports skills performance	Students	Content: Show some interesting jumping methods of pattern rope skipping, and organize students to participate in group games. Frequency: Only once before the intervention. Time: 1 h.	Members of the project team
	Exercise Skills Practice Intervention	Students	Content: Learn 24 kinds of pattern rope skipping movements, including individual patterns, double cooperation and multi-person fun rope skipping. Frequency: one intervention every two weeks, eight times in the last semester, and eight times in the next semester. Time: Each intervention lasts for 45 min.	Teachers, volunteers, and project team members
	Exercise Health Theory Intervention	Students, Primary caregivers	Content: Learn an overview of different sports, the benefits of sports, and learn how to deal with sports injuries. Frequency: one intervention every two weeks, eight times in the last semester, and eight times in the next semester. Time: Each intervention lasts for 45 min.	Teachers, volunteers, and project team members
	Sports Competitions	Students	Content: Arrange eight synchronized rope skipping competitions. Students can choose to participate in different forms of competitions according to their wishes. Frequency: One intervention every four weeks, four times in the last semester, and four times in the next semester. Time: Each intervention lasts for 45 min.	Teachers, volunteers, and project team members
	SP Toolkit	Teachers, Volunteers	Content: Materials, tools, teaching materials, certificates.	Teachers

**Table 2 ijerph-19-14096-t002:** Outcome measurements for the study.

Outcomes	Time			Equipment (Model Number, Manufacturer)	Method
	Baseline	6 Months after Baseline	12 Months after Baseline		
**Anthropometric measures**					
Height	√	√	√	GMCS-I (Xindong Huateng)	Measured to the nearest 0.1 cm at least twice
Weight	√	√	√	RGT-140 (Weighing)	Measured to the nearest 0.1 kg at least twice
Waistline	√	√	√	Tape (MyoTape)	Measured to the nearest 0.1 cm at least twice
Step number	√	√	√	Huawei Band 6 (Huawei, validity (R = 0.72) and reliability (intraclass correlation coefficient = 0.71)	Wear for one week to calculate the average
Calorie expenditure	√	√	√	Huawei Band 6 (Huawei, validity (R = 0.67) and reliability (intraclass correlation coefficient = 0.86)	Wear for one week to calculate the average
Daily minutes of MVPA	√	√	√	Huawei Band 6 (Huawei, validity (R = 0.65) and reliability (intraclass correlation coefficient = 0.76)	Wear for one week to calculate the average
Daily sleep time	√	√	√	Huawei Band 6 (Huawei, validity (R = 0.85) and reliability (intraclass correlation coefficient = 0.91)	Wear for one week to calculate the average
Daily sedentary time	√	√	√	Huawei Band 6 (Huawei, validity (R = 0.74) and reliability (intraclass correlation coefficient = 0.82)	Wear for one week to calculate the average
**PA and other behavioral conditions**					
PA frequency and attitudes of children.	√	√		6 to 17-year-old Chinese children physical activity questionnaire, CCPAQ (The CCPAQ is a valid (R = 0.78) and trustworthy (intraclass correlation coefficient = 0.70) device for measuring PA)	Students should finish the questionnaires in the classroom in the presence of the trained outcome assessors, who can provide guidance and help.
Daily Screen time	√	√		6 to 17-year-old Chinese children physical activity questionnaire, CCPAQ (The CCPAQ is a valid (R = 0.78) and trustworthy (intraclass correlation coefficient = 0.70) device for measuring PA)	Students should finish the questionnaires in the classroom in the presence of the trained outcome assessors, who can provide guidance and help.
**Dietary behavior**					
Daily intake of FV and food intake frequency	√	√		The Food Frequency Questionnaires, FFQ required for the study was developed using the food frequency questionnaire compiled by the 2016 Chinese Resident Dietary Survey with appropriate adjustments according to the study content and purposes of the study, with Cronbach’s > 0.7.	The questionnaires should be self-reported by parents or other primary caregivers of the students.
Children’s FV preferences, identification and the attitude and willingness to eat FV	√	√		Using the Dutch Eating Behavior Questionnaire child-version, DEBQ-C.Zhao Yan et al.developed the Chinese version of the children’s dietary behavior scale according to the English version of DEBQ-C, with Cronbach’s > 0.7	Students should finish the questionnaires in the classroom in the presence of the trained outcome assessors, who can provide guidance and help.
Parental attitudes towards children’s FV intake and SP	√	√		The questionnaire will be adapted from the nutrition knowledge, attitude and praxis questionnaire (KAP) to evaluate children’s knowledge, attitudes and praxis about FV, with Cronbach’s> 0.7.	Students should finish the questionnaires in the classroom in the presence of the trained outcome assessors, who can provide guidance and help.
**Questions and recommendations on the implementation of the interventions**		√		Interview questions included school climate and barriers to implementation of interventions, perceived benefits and challenges to intervention, opinions and perceptions about the intervention’s effects on student involvement, behaviorand class-related outcomes, beliefs about parental involvement, potential for sustainability, overall recommendations to other school principals and ideas for improvements.	The questionnaires should be filled by the trained investigators after face-to-face interviews with school principals, teachers and volunteers.

## Data Availability

Not applicable.

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
