# Peer review of "School Gardening, Cooking and Sports Participation Intervention to Improve Fruits and Vegetables Intake and Moderate-to-Vigorous Physical Activity among Chinese Children: Study Protocol for a Cluster Randomized Controlled Trial"

_ijerph, 2022, doi:10.3390/ijerph192114096_

Round 1

Reviewer 1 Report

This paper describes a study protocol. The aim of the study is to develop and evaluate the effectiveness of school-based gardening and cooking and sports participation interventions on fruits and vegetables intake and moderate-to-vigorous physical activity among Chinese children. The suggestions below should help the authors improve the paper and trial. 

The paper needs careful proof-reading for English language writing and grammatical. E.g., 

Title – should be – “School Gardening, Cooking and Sports Participation Intervention to Improve Fruit and Vegetable Intake …”

Abstract, line 17 – “has become a challenge of global public health problem.” Should be “has become a global public health problem.”

Introduction, line 33 – “An inadequate intake of fruits and vegetables (FV) among children, as well as their 33 insufficient physical activity has become a global health concern[1-3].” Should be “An inadequate intake of fruits and vegetables (FV) and insufficient physical activity among children have become global health concerns [1-3].”

Introduction, line 38 – “Additionally, insufficient moderate to vigorous physical activity (MVPA) is associated with an increased risk of obesity, metabolic syndrome and mental health problems in children [10-12].”

Comments

Abstract

“Primary outcomes include children's FV intake and length of MVPA” – Is this children's average FV intake and duration (use duration instead of length) of MVPA per day or per week? I.e., is the terms “average daily” or “average weekly”

Keywords

-       These should not duplicate words in the title. 

Introduction

-       “in particular that multifaceted interventions using both diet and sports participation (SP) are more effective in improving children's health[15-17],”- More effective than what? Diet or physical activity interventions alone? Need to specify this. 

-       “Most of the current diet and SP program for children in China have been designed 45 with the specific goal of improving obesity, neglecting the development of lifestyle-related 46 interventions such as improving children's eating and SP behaviour[18-20].” This seems strange as most obesity treatment/prevention interventions target diet and physical activity. Can you further explain how programs with “the specific goal of improving obesity” neglect children’s eating and physical activity behaviours?

-       Throughout the introduction it appears that the term SP (Sports participation) is used interchangeably with (or instead of) physical activity. Physical activity is any bodily movement that results in energy expenditure. Sport is a particular type of physical activity with rules and, typically, competition. Often lifestyle interventions promote physical activity rather than just sport. Please check throughout and modify SP to PA where appropriate. E.g., “neglecting the development of lifestyle-related interventions such as improving .. SP behaviour[18-20]” Should this PA behaviour?

-       “Therefore, it is essential to design a sustain-75 able and structured intervention program for school-age children based on SGC and SP 76 intervention.” – This statement could be improved – “Therefore, implementing a structured school-based intervention program for children based on SGC and SP aligns with curriculum requirements in China.”

-       In the introduction, can similar previous trials conducted in China be briefly described, and the gap in knowledge identified that the proposed trial will fill. Specifically, what were the results of the below trial, how is the proposed trial different, and what gap will be filled by the proposed trial? 

Liu, Z.; Gao, P.; Gao, A.-Y.; Lin, Y.; Feng, X.-X.; Zhang, F.; Xu, L.-Q.; Niu, W.-Y.; Fang, H.; Zhou, S., et al. Effectiveness of a Multifaceted Intervention for Prevention of Obesity in Primary School Children in China: A Cluster Randomized Clinical Trial. Jama Pediatr 2022176, e214375-e214375, doi:10.1001/jamapediatrics.2021.4375.

-        

Methods

-       From Figure 1, is it correct to say that the SP&SGC intervention group gets double the intervention time of the other groups (2 x 90min/week vs 1 x 90min/week). Could that be stated clearly in the methods. 

-       Figure 2 is providing limited information about how SCT is applied in the intervention. It would be more useful to explain how the intervention targets and addresses key constructs of the theory, e.g., in table 1. 

-       In relation to the comments in the introduction about PA and SB, is it more appropriate to call the movement component of the intervention a “SP” intervention or a “Physical Activity” intervention?

-       Will assessors be blinded to intervention groups?

-       The detail about the measures is limited as it is only provided in table 2 and is separate to the outcomes (2.9). Table 2 needs to identify how each outcome in 2.9 is being assessed (eg by including the outcomes). For example, how is “ Daily minutes of MVPA (unit: minutes).” being assessed – it could be by the Huawei Band 6 or the Chinese children physical activity ques-tionnaire, CCPAQ. This is particularly important for the primary outcomes. 

-       Related to the above point, details need to be provided for all measures of the validity and reliability of the measure in the age group. For example, what is the validity and reliability of the Huawei Band 6 for measuring steps, calories, sleep and Length of medium and high in-tensity physical activity in children?

-       Statistical analyses: This is a cluster randomized trial but the analysis plan doesn’t appear to account for the clustered nature of the data (children are recruited from within classes within schools within districts). This needs to be reviewed. 

Discussion

-       The discussion is only 2 paragraphs long but the second paragraph repeats strengths listed in the first. Please remove the repeated information and consider further what can be included in the discussion section. (Second, the implementation of the intervention involves not only students 310 but also primary caregivers and teachers to increase the potential for sustainability. 311 Third, this study will assess changes in outcomes using both self-reported and objective 312 measures, with more rigorous results. Fourth, the study will develop a process evalua-313 tion plan for the implementation of the intervention). 

Author Response

Reviewer 1

Point 1: The paper needs careful proof-reading for English language writing and grammatical. E.g., 

Response: We had assigned one of our corresponding author, a native-speaker, to polish the language of the study and correct some mistakes in language use.

Point 2: Title – should be – “School Gardening, Cooking and Sports Participation Intervention to Improve Fruit and Vegetable Intake …”

Response: Thank you for this suggestion. We have changed the title to "School Gardening, Cooking and Sports Participation Intervention to Improve Fruits and Vegetables Intake and Moderate-to-Vigorous Physical Activity among Chinese Children: Study Protocol for a Cluster Randomised Controlled Trial".

Point 3: Abstract, line 17 – “has become a challenge of global public health problem.” Should be “has become a global public health problem.”

Response: Thank you. We have revised it.

Point 4: Introduction, line 33 – “An inadequate intake of fruits and vegetables (FV) among children, as well as their 33 insufficient physical activity has become a global health concern[1-3].” Should be “An inadequate intake of fruits and vegetables (FV) and insufficient physical activity among children have become global health concerns [1-3].”

Response: Thank you. We have revised it.

Point 5: Introduction, line 38 – “Additionally, insufficient moderate to vigorous physical activity (MVPA) is associated with an increased risk of obesity, metabolic syndrome and mental health problems in children [10-12].”

Response: Thank you. We have revised it.

Point 6: Abstract

“Primary outcomes include children's FV intake and length of MVPA” – Is this children's average FV intake and duration (use duration instead of length) of MVPA per day or per week? I.e., is the terms “average daily” or “average weekly”

Response: Thank you for this suggestion. We had used duration instead of length and added the terms “per day”.

Point 7: Keywords

-       These should not duplicate words in the title. 

Response: Thank you for this suggestion. We had revised it.

Point 8: Introduction

-       “in particular that multifaceted interventions using both diet and sports participation (SP) are more effective in improving children's health[15-17],”- More effective than what? Diet or physical activity interventions alone? Need to specify this. 

Response: Thank you for this suggestion. We had changed it to  “in particular that multifaceted interventions using both diet and sports participation (SP) are more effective than diet or SP interventions alone in improving children's health.

Point 9: Introduction

-      “Most of the current diet and SP program for children in China have been designed 45 with the specific goal of improving obesity, neglecting the development of lifestyle-related 46 interventions such as improving children's eating and SP behaviour[18-20].” This seems strange as most obesity treatment/prevention interventions target diet and physical activity. Can you further explain how programs with “the specific goal of improving obesity” neglect children’s eating and physical activity behaviours?

Response: Thank you for this suggestion. What we actually mean is that many current diet and SP programmes, such as free FV distribution to children and high intensity intermittent exercise for children, do improve childhood obesity but are not sustainable, only through interventions to improve children's lifestyles can their health status be completely addressed.

Point 10: Introduction

-       Throughout the introduction it appears that the term SP (Sports participation) is used interchangeably with (or instead of) physical activity. Physical activity is any bodily movement that results in energy expenditure. Sport is a particular type of physical activity with rules and, typically, competition. Often lifestyle interventions promote physical activity rather than just sport. Please check throughout and modify SP to PA where appropriate. E.g., “neglecting the development of lifestyle-related interventions such as improving .. SP behaviour[18-20]” Should this PA behaviour?

Response: Thank you for this suggestion. It should be PA behaviour, we had revised all of them in line 50 and 54.

Point 11: Introduction

-       “Therefore, it is essential to design a sustain-75 able and structured intervention program for school-age children based on SGC and SP 76 intervention.” – This statement could be improved – “Therefore, implementing a structured school-based intervention program for children based on SGC and SP aligns with curriculum requirements in China.”

Response: Thank you for this suggestion. We had revised it.

Point 12: Introduction

-       In the introduction, can similar previous trials conducted in China be briefly described, and the gap in knowledge identified that the proposed trial will fill. Specifically, what were the results of the below trial, how is the proposed trial different, and what gap will be filled by the proposed trial? 

Liu, Z.; Gao, P.; Gao, A.-Y.; Lin, Y.; Feng, X.-X.; Zhang, F.; Xu, L.-Q.; Niu, W.-Y.; Fang, H.; Zhou, S., et al. Effectiveness of a Multifaceted Intervention for Prevention of Obesity in Primary School Children in China: A Cluster Randomized Clinical Trial. Jama Pediatr 2022, 176, e214375-e214375, doi:10.1001/jamapediatrics.2021.4375.

Response: Thank you for this suggestion. We had added this part, “A recent study used the multifaceted intervention effectively reduced the mean BMI and obesity prevalence in primary school children across socioeconomically distinct regions in China, but it did not change MVPA, physical fitness and other outcomes. Mainly because the researchers did not use sport participation interventions that were fun and challenging, instead, students are required to do PA for a total of one hour per day, which does not allow for a good level of intensity and duration of PA for children. Therefore, the choice of a sensible intervention is crucial.”

Point 13: Methods

-       From Figure 1, is it correct to say that the SP&SGC intervention group gets double the intervention time of the other groups (2 x 90min/week vs 1 x 90min/week). Could that be stated clearly in the methods. 

Response: Thank you for this suggestion. In fact, the SP&SGC intervention group, the SP intervention group, and the regular practice group received 180 minutes(4 x 45min/week) of intervention per week. We have added the explanation to the methods section.

Point 14: Methods

-       Figure 2 is providing limited information about how SCT is applied in the intervention. It would be more useful to explain how the intervention targets and addresses key constructs of the theory, e.g., in table 1. 

Response: Thank you for this suggestion. We had added more information in Figure 2, please check it.

Point 15: Methods

-       In relation to the comments in the introduction about PA and SB, is it more appropriate to call the movement component of the intervention a “SP” intervention or a “Physical Activity” intervention?

Response: Thank you for this suggestion. It would indeed be more appropriate to use SP intervention instead of the movement component of the intervention, we had revised it.

Point 16: Methods

-       Will assessors be blinded to intervention groups?

Response: Thank you for this question. It is unlikely that data collectors will be blinded because of the nature of the study. But data analysts will be blinded. We have added two sentences to explain blinding (line 149-151).

  “Blinding of the school allocation in this setting will not be possible. Therefore, the investigators and data collectors will not be blinded in this trial. However, the outcome assessors and the data analysts will be blinded.”

Point 17: Methods

-       The detail about the measures is limited as it is only provided in table 2 and is separate to the outcomes (2.9). Table 2 needs to identify how each outcome in 2.9 is being assessed (eg by including the outcomes). For example, how is “ Daily minutes of MVPA (unit: minutes).” being assessed – it could be by the Huawei Band 6 or the Chinese children physical activity ques-tionnaire, CCPAQ. This is particularly important for the primary outcomes. 

Response: Thank you for this suggestion. We had corrected and marked them all in the Table 2 and (2.9).

Point 18: Methods

-       Related to the above point, details need to be provided for all measures of the validity and reliability of the measure in the age group. For example, what is the validity and reliability of the Huawei Band 6 for measuring steps, calories, sleep and Length of medium and high in-tensity physical activity in children?

Response: Thank you for this suggestion. We had corrected and marked them all in the Table 2 and (2.9).

Point 19: Methods

-       Statistical analyses: This is a cluster randomized trial but the analysis plan doesn’t appear to account for the clustered nature of the data (children are recruited from within classes within schools within districts). This needs to be reviewed. 

Response: We agree with reviewer’s opinion. We revised the section 2.2.1 “recruitment of the schools”, have added the criteria for schools in (line 114-121).

“The schools will be randomly selected if they meet the criteria described below:

Inclusion criteria: The schools will be included if they were public primary schools and with a minimum of 300 primary students. The school principal agrees to the randomization process and comply with this study protocol.

Exclusion criteria: The schools are boarding schools and schools for special populations, as are schools with fewer than two classes in grades 4-5, or schools with plans to cease operating within a year, and schools without established gardens or sufficient space to plan garden construction, or receiving other similar interventions will be excluded. “

Point 20: Discussion

-       The discussion is only 2 paragraphs long but the second paragraph repeats strengths listed in the first. Please remove the repeated information and consider further what can be included in the discussion section. (Second, the implementation of the intervention involves not only students 310 but also primary caregivers and teachers to increase the potential for sustainability. 311 Third, this study will assess changes in outcomes using both self-reported and objective 312 measures, with more rigorous results. Fourth, the study will develop a process evalua-313 tion plan for the implementation of the intervention). 

Response: Thank you for this suggestion. In response to your suggestions we have removed the repetitive discussion and added a discussion on the sustainability of the project.

Reviewer 2 Report

The paper describes the protocol for a cluster randomised controlled trial to investigate whether a gardening, cooking and sports participation intervention in school improves the intakes of fruits and vegetables and increases moderate-vigorous physical activity among 4-5 grade children. The protocol is well written and the specific evaluation methods are described clearly. I do have some comments for the authors outlined below.

General comments:

·       Include the age of the children, perhaps in the intro (are they 9-11 years?) Useful for international reader

·         Could the authors comment on or acknowledge the potential differences between the schools in terms of the facilities available? Prior work in secondary schools shows us that schools can differ greatly in terms of space for garden and sports, teaching capacity, cooking facilities etc, which could affect the delivery of the intervention and, ultimately, the results.

·         Recruitment of the School Children: Will the children who are ineligible to take part still have access to the intervention? I.e. will they will still attend the lessons with their peers? I understand those with medical conditions may not be able to participate in the PA, but they could still attend the SGC aspects even if their data is not collected?

·         Please acknowledge the lack of an objective measure of f&v intake, for example the use of serum carotenoid as a dietary biomarker.

·         FFQs are completed by the parent, which, considering the age group of students, is a suitable approach. Has the FFQ been validated in terms of responsiveness? How accurately can the FFQ tool measure intervention effects, i.e. is it sensitive enough to detect changes in dietary intake? This is particularly important in an intervention context.

Line 84-85: Please specify what the ‘other healthy conditions’ are.

Figure 1: I find this figure confusing. I wonder whether the authors could revise the layout of the flow chart. Perhaps flowing vertical rather than horizontal? Maybe some indication of timing/time points might help.

Table 2: Rename ‘calorie expenditure’ in anthropometric measures, to be more specific.

Table 2: Should ‘dietary behaviour be highlighted? Is it a sub-heading? To make the table clearer, perhaps use lines to separate the different outcome categories; anthropometric, PA, dietary behaviour, effectiveness of implementation.

Table 2: ‘Children should eat about fruits and vegetables’ does not make sense.

Section 2.9: Some of the outcomes have gone across 2 lines (2 numbers). Lines 253-4, 259-260 and 263-4.

Author Response

Reviewer 2

Point 1: ·       Include the age of the children, perhaps in the intro (are they 9-11 years?) Useful for international reader

Response: Thank you for this suggestion. We had added the age of children in the introduction( line 84 ).

Point 2: ·         Could the authors comment on or acknowledge the potential differences between the schools in terms of the facilities available? Prior work in secondary schools shows us that schools can differ greatly in terms of space for garden and sports, teaching capacity, cooking facilities etc, which could affect the delivery of the intervention and, ultimately, the results.

Response: Thank you for this suggestion. Considering the potential differences in available facilities between schools, we have designed and patented a three-dimensional, movable mini-garden for use in schools, which has a small footprint and can follow the light. The main form of cooking intervention is the preparation of fruit and vegetable salads, which do not require complicated cooking tools and can be done in the classroom. The other main form of SP intervention is rope skipping, a sport that requires little equipment or space and is easy to implement in schools. We can therefore try to ensure that the school's gardening, cooking and sports participation intervention facilities are as uniform as possible.

Point 3: ·         Recruitment of the School Children: Will the children who are ineligible to take part still have access to the intervention? I.e. will they will still attend the lessons with their peers? I understand those with medical conditions may not be able to participate in the PA, but they could still attend the SGC aspects even if their data is not collected?

Response: Thank you for this question. In fact SGC interventions also involve PA such as watering and fertilising, so in principle children with medical conditions are not allowed to participate, but on your suggestion we will consider allowing these children to participate in some of these activities.

Point 4: ·         Please acknowledge the lack of an objective measure of f&v intake, for example the use of serum carotenoid as a dietary biomarker.

Response: Thank you very much for your suggestion and I have to say that the use of serum carotenoid as a dietary biomarker in future studies is a good practice and we have added that limitation to the discussion.

Point 5: ·         FFQs are completed by the parent, which, considering the age group of students, is a suitable approach. Has the FFQ been validated in terms of responsiveness? How accurately can the FFQ tool measure intervention effects, i.e. is it sensitive enough to detect changes in dietary intake? This is particularly important in an intervention context.

Response: Thank you for pointing out this. The FFQ was developed based on the 2016 Chinese Resident Dietary Survey and the 2017 Chinese Residents' Nutrition and Health Status Monitoring, and it has been adapted, and validated in the preliminary study (Cronbach’s α coefficient >0.7). The information can be found in Table 2.

Point 6: Line 84-85: Please specify what the other healthy conditions’ are.

Response: Thank you for this suggestion. We describe other health conditions in detail.

Point 7: Figure 1: I find this figure confusing. I wonder whether the authors could revise the layout of the flow chart. Perhaps flowing vertical rather than horizontal? Maybe some indication of timing/time points might help.

Response: Thank you for this suggestion. We had changed it to a flow chart and added the time points, it indeed looked more clearly than before.

Point 8: Table 2: Rename ‘calorie expenditure’ in anthropometric measures, to be more specific.

Response: Thank you. We have revised it.

Point 9: Table 2: Should‘dietary behaviour be highlighted? Is it a sub-heading? To make the table clearer, perhaps use lines to separate the different outcome categories; anthropometric, PA, dietary behaviour, effectiveness of implementation.

Response: The dietary behaviour indeed should be highlighted. Thank you for your suggestion. We had use lines to separate the different outcome categories to make the table clearer.

Point 10: Table 2: ‘Children should eat about fruits and vegetables’ does not make sense.

Response: Thank you for this suggestion. We had changed it to “Children's FV preferences, identification and the attitude and willingness to eat FV”.

Point 11: Section 2.9: Some of the outcomes have gone across 2 lines (2 numbers). Lines 253-4, 259-260 and 263-4.

Response: Thank you for this suggestion. We had revised all of them.

Reviewer 3 Report

It is an interesting research protocol, and the aim is to develop and evaluate the effectiveness of SGC and SP interventions on FV intake and MVPA. However, it is not clear what the scientific problem of this study is, so the relationship between the research method and the research outcome variable is inconsistent, and the manuscript is recommended to be major revised. There are also several awkward sentences that authors could reconsider.

1. LINE 26 “length of MVPA” is the same as Time of MVPA”?

2. Figure 1. Is the baseline survey primary “caretakers” caregivers?

3. Figure 1. The intervention of the control group uses 45 minutes/time class, but SP and SGC groups use 90 minutes/time for an intervention. Is it appropriate for elementary school students to do 90 minutes of continuous activity?

4. Figure 1. One week only one time for the SP group is appropriate for “one hour of school exercise per day”?

5. LINE 133-139 Sample Size Estimation is not clear. What is the mean of “60min p” and “week p”?

6. Table 2  “medium and high” is the same as “moderate-to-vigorous” in “Length of medium and high intensity physical activity”?

7. Table 2  “7 to 17-year-old” is the same as “6 to 17-year-old”?

8. LINE 254 and LINE 260 can not be the independent outcome

9. 2.11. Cost-Effectiveness Analysis, the calculation method is unclear. What is the feasibility and necessity of its investigation? Please explain. 

Author Response

Reviewer 3

Point 1: 1. LINE 26 “length of MVPA” is the same as “Time of MVPA”?

Response: Yes it is, one of the reviewers suggested we change to a uniform “duration of MVPA”, we had revised all of them, thanks for your remind.

Point 2: 2. Figure 1. Is the baseline survey primary “caretakers” caregivers?

Response: Thanks for your remind, we had revised it.

Point 3:  Figure 1. The intervention of the control group uses 45 minutes/time class, but SP and SGC groups use 90 minutes/time for an intervention. Is it appropriate for elementary school students to do 90 minutes of continuous activity?

Response: Thank you for this suggestion. In fact, the SP&SGC intervention group, the SP intervention group, and the regular practice group received 180 minutes(4 x 45min/week) of intervention per week. We had revised related content in Figure 1.

Point 4: Figure 1. One week only one time for the SP group is appropriate for “one hour of school exercise per day”?

Response: In fact, Chinese primary schools have a sports club every day and we use two of the days for each 45-minute SP intervention, teaching children to master fun and challenging sports skills, after which the children are given PA tasks to ensure that they can achieve an MVPA of one hour every day.

Point 5:  LINE 133-139 Sample Size Estimation is not clear. What is the mean of “60min p” and “week p”?

Response: Thanks for your remind. Actually “60min p”means “60min per day”, and should revise “per day” instead of “week p”. we had revised them. And we recalculated the sample size after modifying, please check it.

Point 6: Table 2  “medium and high” is the same as “moderate-to-vigorous” in “Length of medium and high intensity physical activity”?

Response: Thank you for this remind. The “medium and high” is the same as “moderate-to-vigorous”, and we had revised it in Table 2.

Point 7: Table 2  “7 to 17-year-old” is the same as “6 to 17-year-old”?

Response: It’s the same. We apologise that there is an error in the display due to a formatting issue, and we had revised it, thanks for your remind.

Point 8: LINE 254 and LINE 260 can not be the independent outcome

Response: We apologise that there is also an error in the display due to a formatting issue, and we had revised it, thanks for your remind.

Point 9: 2.11. Cost-Effectiveness Analysis, the calculation method is unclear. What is the feasibility and necessity of its investigation? Please explain.

Response: Thank you for the suggestion. We now have revised the section 2.11. “Cost-Effectiveness Analysis” as following:

 We will assess the cost-effectiveness of the program in terms of cost per case of fruit and vegetable intake or MVPA improved.

Cost: The costs will include the time spent by project staff, school staff and the student's primary caregiver (in most cases the parent) for all intervention activities and material costs. Only the time spent by project staff implementing the intervention will be included. Time costs will be based on individual employment compensation (if available) or the average compensation for similar employees in the local area. The cost of garden materials, seeds, fertilizer, kitchen utensils, kettles and hoes are subject to actual purchase prices.

Effectiveness: The impact of the program will be measured by changes in fruit and vegetable intake or MVPA. The cost-effectiveness ratio will be calculated as follows:

  1. Cost of Intervention Services for one primary school child.
  2. Cost for each case of improved fruit and vegetable intake.
  3. Cost for each case of improved MVPA.

Round 2

Reviewer 3 Report

Dear authors,

Thank you for considering my suggestions and incorporating them into the manuscript.

I hope you can  successfully complete this protocol.